# BRAF Inhibitors in Metastatic Colorectal Cancer and Mechanisms of Resistance: A Review of the Literature

**DOI:** 10.3390/cancers15215243

**Published:** 2023-10-31

**Authors:** Patricia Guerrero, Víctor Albarrán, María San Román, Carlos González-Merino, Coral García de Quevedo, Jaime Moreno, Juan Carlos Calvo, Guillermo González, Inmaculada Orejana, Jesús Chamorro, Íñigo Martínez-Delfrade, Blanca Morón, Belén de Frutos, María Reyes Ferreiro

**Affiliations:** Department of Medical Oncology, Ramon y Cajal University Hospital, 28034 Madrid, Spain; valbarran@salud.madrid.org (V.A.); mariavictoria.san@salud.madrid.org (M.S.R.); cgmerino@salud.madrid.org (C.G.-M.); cgquevedosuero@salud.madrid.org (C.G.d.Q.); jmdoval@salud.madrid.org (J.M.); jcalvop@salud.madrid.org (J.C.C.); guillermo.gonzalez.martin@salud.madrid.org (G.G.); inmaculada.orejana@salud.madrid.org (I.O.); jchamorro@salud.madrid.org (J.C.); imdelfrade@salud.madrid.org (Í.M.-D.); blancaisabel.moron@salud.madrid.org (B.M.); bfrutos@salud.madrid.org (B.d.F.); mariareyes.ferreiro@salud.madrid.org (M.R.F.)

**Keywords:** BRAF, inhibitors, resistance, colorectal cancer

## Abstract

**Simple Summary:**

The MAP kinases pathway has shown a key role in the pathogenesis of colorectal cancer. The development of targeted therapy against BRAF-mutated tumors is changing the management of advanced disease. A proper understanding of the mechanisms of acquired resistance is essential to optimize the results of systemic treatment. We aim to review the current knowledge and potential fields of research regarding the use of BRAF inhibitors in metastatic colorectal tumors.

**Abstract:**

Metastatic colorectal cancer (mCRC) with mutated BRAF exhibits distinct biological and molecular features that set it apart from other subtypes of CRC. Current standard treatment for these tumors involves a combination of chemotherapy (CT) and VEGF inhibitors. Recently, targeted therapy against BRAF and immunotherapy (IT) for cases with microsatellite instability (MSI) have been integrated into clinical practice. While targeted therapy has shown promising results, resistance to treatment eventually develops in a significant portion of responsive patients. This article aims to review the available literature on mechanisms of resistance to BRAF inhibitors (BRAFis) and potential therapeutic strategies to overcome them.

## 1. Introduction

CRC is currently one of the most common neoplasms, being the third most frequent tumor in men and the second in women [1]. Its incidence is rising in Western countries, with up to 70% of CRC being sporadic and associated with environmental factors, like tobacco, alcohol, diet, sedentary lifestyle, or obesity [2]. Globally, CRC accounts for 11% of all new cancer cases worldwide [3], with up to a quarter presenting distant metastases at diagnosis. Among the remaining three-quarters eligible for surgery, nearly half will develop metastases during the course of the disease [2]. Hence, this poses a significant healthcare challenge.

The *BRAF* proto-oncogene is located on chromosome 7 and consists of 18 exons [2]. Mutation in this gene is present in 7% of all tumors, with the majority being melanomas, where *BRAF* mutation is observed in up to 50% [4]. In mCRC, the *BRAF* mutation frequency hovers around 12% [5]. *BRAF* mutation in mCRC carries independent prognostic value and significant associations with clinical and biological disease characteristics [6]. This has led to a growing interest in genetic testing for *BRAF* mutations and the development of targeted therapies against them. 

## 2. Molecular Biology

### 2.1. Carcinogenesis

The carcinogenesis of colorectal neoplasias starts with the accumulation of various genetic and epigenetic mutations that transform healthy epithelial tissue into benign neoplasms (adenomatous polyps), which later progress to dysplastic polyps and eventually invasive carcinomas. *BRAF* mutation is believed to be involved in the transformation of healthy epithelium into serrated adenomas (sessile or traditional), potentially representing an early event in CRC progression [7]. There are two main pathways through which adenomatous polyps transform into CRC: microsatellite instability (MSI) and chromosomal instability (CIN) [2]. Most CRCs develop through the latter pathway. One of the most frequently implicated genes is *APC*, which forms the beta-catenin destruction complex that activates the Wnt signaling pathway. Alteration of the *APC* gene (most commonly a 5q deletion) inhibits beta-catenin destruction, resulting in hyperactivation of the Wnt signaling pathway, leading to increased cellular proliferation and invasion. In terms of MSI tumors, they result from both the mismatch repair pathway (MMR) and the hypermethylated CpG island phenotype (CIMP) [2]. In the MMR pathway, mutations occur in genes encoding DNA repair proteins (*MLH-1*, *MSH2*, *MSH-6*, *PMS-2*), leading to progressive accumulation of unrepaired DNA abnormalities. These DNA molecules code for truncated proteins that serve as neoantigens, making these tumors more immunogenic than others. On the other hand, the CIMP pathway involves hypermethylation of the repair protein genes, specifically, CpG islands in the MMR enzyme promoter region. This extensive methylation inactivates repair proteins, resulting in epigenetic instability. Sporadic microsatellite instability is associated with a higher prevalence of *BRAF* mutations compared to tumors with microsatellite stability (MSS) or non-polyposis hereditary CRC [8]. In sporadic CRC, *BRAF* mutation is observed in 60% of tumors with MSI, whereas it is seen in only 5–10% of MSS tumors [1]. Approximately 20% of patients with mutated *BRAF* V600E concurrently exhibit microsatellite instability [9].

### 2.2. BRAF Mutation

BRAF is a serine/threonine kinase belonging to the RAF family, crucial in the Mitogen-Activated Protein Kinase (MAPK) signaling pathway, also known as the RAS/RAF/MEK/ERK cascade. This pathway controls cell growth and proliferation in response to various growth factors [5].

Different subtypes of RAF protein exist: A, B, and C, with subtype B (BRAF) being the most significant player in MEK and exhibiting more mutations. It comprises three domains: CR1 and CR2 located in the N-terminal region, where CR1 is the binding domain and CR2 is the regulatory domain. The third domain, CR3, is situated in the C-terminal region and is the catalytic domain [10]. Apart from its role in the MAPK pathway, BRAF is involved in other cellular processes, like migration (via RHO small GTPases), apoptosis (via BCL-2 regulation), and survival (via the HIPPO pathway) [7]. *BRAF* mutations are classified into three classes, class I being the most frequent mutation type (representing 95%) [2], occurring in codon 600, where valine is changed to glutamine (*BRAF* V600E). Although this mutation is present in other neoplastic diseases, CRC is unique in its description of glutamic acid substitution at codon 600, while other tumors more commonly feature lysine or methionine substitutions [11]. Class II *BRAF* mutations involve mutations in codons 601/597 and exhibit a prognosis similar to class I mutations. Finally, class III mutations (like *BRAF* D594G or G596N) act by amplifying *RAS* or *CRAF*. In class I and II mutations, the protein escapes the negative feedback that would normally inhibit its signaling under physiological conditions, leading to permanent activation. However, class III mutations require a second hit to be oncogenic, such as *KRAS* mutation, *NRAS* mutation, or EGFR receptor activation [12]. Furthermore, class III mutations impair the kinase (kinase-impairing mutation), while V600E mutations activate the kinase (kinase-activating mutation), possibly explaining the better prognosis of the former: Mutations like D594G or G596N exhibit an overall survival (OS) of 60.7 months, significantly longer than the 11.4 months seen in *BRAF* V600E mCRC or the 43 months in *BRAF* wild-type subtypes [13]. While class II mutations coexist with other pathway mutations in 10–20% of cases, class I *BRAF* mutation is almost mutually exclusive with other MAPK pathway mutations [14], with coexistence of *RAF* and *RAS* mutations extremely rare. In a report involving 2530 patients from three randomized trials (COIN, PICCOLO, and FOCUS), such coexistence was observed in only eight cases (0.3%) [15]. Furthermore, there is a subclassification of the different *BRAF* mutations, as mentioned earlier, which also implies prognostic differences:-*BRAF* V600E: In general terms, it confers a worse prognosis, with an overall survival between 9 and 19 months. In the CALGB 80405 first-line trial, patients with the BRAF V600E mutation were administered FOLFOX or FOLFIRI plus Bevacizumab or Cetuximab compared to those with *BRAF* wild type. An average overall survival of 13.5 months was observed compared to 30.6 months in the *BRAF* wild-type patients (*p* = 0.001) [16]. They exhibit a poorer response to chemotherapy, as well as a higher incidence of peritoneal metastases, which results in symptomatic issues significantly affecting the patient’s quality of life (abdominal pain, ascites, intestinal or urinary obstruction, etc.) [17]. Regarding pathological anatomy, they often present mucinous histology and are poorly differentiated. They are more commonly associated with a specific phenotype: women, advanced age, and tumors preferentially located in the right colon. As mentioned earlier, at the molecular level, it is frequent to observe *KRAS* wild-type tumors, and there is a higher coexistence with MSI. In a study involving 3063 patients, 35% of dMMR CCRm had a concurrent *BRAF* V600E mutation, while only 7% of pMMR tumors had this mutation (*p* < 0.001) [9].-*BRAF* non-V600E: These are more frequently observed in men, at younger ages than *BRAF* V600E mutations, and they are more commonly located in the left colon. They usually coexist with mutations in *KRAS*, and there is a lower association with MSI.

Another subclassification of tumors with *BRAF* mutations is based on genetic expression:-BM1: This is the most frequent (observed in up to 70%). It is defined by the activation of the KRAS/AKT pathway, dysregulation of mTOR/4EBP, and increased immune infiltration [18]. Generally, it confers a worse prognosis, although it exhibits a greater response to BRAF, MEK, and EGFR inhibitors compared to BMS-2 [19].-BM2: This represents the remaining 30%, and it is characterized by the dysregulation of cell cycle checkpoints [18].

A classification into four molecular subgroups (CMS 1–4) has also been developed, which has a significant association with the clinical and biological characteristics of the tumor [15]. It is worth noting that up to 60–70% of CRC tumors with mutated *BRAF* belong to the CMS1 subtype, also known as immune MSI, which is characterized by a high mutation rate, but a low prevalence of somatic copy number alterations (SCNAs). These tumors encompass those with MSI, which could explain the success of the combination of BRAF inhibitors and immunotherapy [20,21].

## 3. Mechanism of Action of BRAF Inhibitors (BRAFis)

There is a wide variety of BRAFis, all of which have a similar mechanism of action that ultimately inhibits downstream signaling of BRAF within the MAPK pathway. Some of the most commonly used ones are vemurafenib, dabrafenib, and encorafenib [22]. They selectively and competitively inhibit the mutated kinase, targeting the ATP-binding domain of the monomeric form of BRAF [23]. Most of them are also active against other forms of RAF, such as CRAF [1]. They are administered orally and are generally well tolerated. All of them can be safely administered in patients with a glomerular filtration rate (GFR) greater than 30 mL/min, and there is currently no safety data available for lower GFRs. Encorafenib has demonstrated a longer pharmacodynamic activity than other BRAF inhibitors. In a study, it was observed that despite the fact that the drug concentrations required to inhibit BRAF V600E were similar among the three drugs, the half-life of encorafenib was significantly longer (30 h) than that of dabrafenib (2 h) and vemurafenib (0.5 h) [24]. The most common adverse effects associated with its administration are palmoplantar erythroderma [24], reported in up to 67% of CCRm patients with *BRAF* V600E mutation treated with encorafenib monotherapy. The most common dose-limiting toxicity was peripheral neuropathy, manifesting as neuralgia (4.1%) [1]. Vemurafenib and dabrafenib have a different toxicity profile compared to encorafenib, as they are more frequently associated with digestive alterations (vomiting or diarrhea), rather than neurological events. 

## 4. Current Treatment Landscape in BRAF-Mutated mCRC

### 4.1. BRAF-Mutated mCRC without MSI: First-Line Systemic Treatment

Current data on *BRAF*-mutated colon tumors have shown worse outcomes compared to *BRAF* wild-type tumors. In the FOCUS study with 711 patients, the median overall survival (OS) after standard treatment was about 12 months, compared to nearly 30 months in patients with *BRAF* wild-type tumors [25]. Despite not observing statistically significant differences in terms of PFS, only 33% of *BRAF*-mutated patients received second-line treatment, compared to 50% in *BRAF* wild-type cases [17]. In an attempt to overcome the poor response to standard chemotherapy, treatment with EGFR inhibitors (EGFRis) was explored as an alternative. However, a meta-analysis concluded that EGFR blockade does not enhance the efficacy of standard chemotherapy in *BRAF*-mutated mCRC [26,27]. Despite promising results initially obtained in the TRIBE study with FOLFOXIRI + Bevacizumab versus FOLFIRI + Bevacizumab in the *BRAF* V600E mutated subgroup (median OS of 19 and 10.7 months, respectively; median PFS of 7.5 and 5.5 months, respectively), subsequent studies showed that the triplet chemotherapy has not shown to be outright superior to doublet regimens, while increasing toxicity [28]. However, the FIRE 4.5 study, which compares FOLFOXIRI + Cetuximab and FOLFOXIRI + Bevacizumab as first-line treatment in CRC with *BRAF* mutated, recommends the use of the triplet + Bevacizumab as the preferred option for patients with ECOG 0-1 [29]. In the ESMO guidelines, this combination is also referenced as the first choice for fit patients [30]. The recent CAIRO [31] study shows positive results in favor of FOLFOXIRI + Bevacizumab vs. FOLFOX/FOLFIRI + Bevacizumab in right-sided CRC patients with initially unresectable hepatic metastases and/or *RAS*/*BRAF* V600E mutation, with a PFS of 10.6 vs. 9 months, ORR of 52.1% vs. 32%, and R0/1 of 51.4% vs. 37.4%, respectively. However, these results were achieved at the expense of increased toxicity, with a rate of G3–G4 events of 75% in patients who received the triplet compared to 58.5% in those who did not.

Therefore, the current standard of care for first-line treatment in non-MSI mCRC with *BRAF* mutations includes standard chemotherapy containing fluoropyrimidines (FOLFOX vs. FOLFIRI) combined with VEGF inhibitors (Bevacizumab) [32], knowing that FOLFOXIRI + Bevacizumab can be administered to selected patients.

### 4.2. BRAF-Mutated mCRC with MSI

As mentioned earlier, MSI tumors are more immunogenic than other tumor subtypes, as the truncated proteins resulting from errors in unrepaired DNA strands function as neoantigens that activate the immune system. The KEYNOTE-177 study demonstrated that pembrolizumab (anti-PD-1) in monotherapy was superior to standard chemotherapy in the first-line setting. Patients receiving immunotherapy had a progression-free survival (PFS) of 16.5 months compared to 8.2 months in the chemotherapy arm (HR: 0.60, *p* = 0.0002). The PFS benefit was also observed in subgroups carrying *BRAF* mutations (HR 0.48; 95% CI 0.27–0.86), while tumors with *KRAS* or *NRAS* mutations did not derive this benefit (HR 1.19, 95% CI 0.68–2.07) [33]. Therefore, having a *BRAF* mutation does not necessarily lead to a worse response to immunotherapy in MSI patients. Immunotherapy’s benefit in MSI tumors has been confirmed in other studies [34,35], making it the current first-line treatment for MSI-positive mCRC. Various studies have been designed to explore the combination of immunotherapy and targeted therapy as a therapeutic option for these tumors. In a phase II trial, dabrafenib, trametinib, and spartalizumab (anti-PD-1) were administered to a total of 21 patients with *BRAF* V600E mutations who had not received prior immunotherapy or BRAFi treatment. An objective response rate (ORR) of 35% and a disease control rate (DCR) of 75% were achieved [36].

### 4.3. BRAF-Mutated mCRC without MSI: Second-Line Systemic Treatment

Due to limited response to first-line treatment in *BRAF*-mutated mCRC, studies were initiated in the second-line setting to assess the efficacy of targeted treatment against this mutation: BRAF inhibitors (BRAFis). Unlike melanoma, BRAFi monotherapy showed limited effectiveness: a phase I study with vemurafenib (PLX4032) achieved a partial response (PR) of 5% and a PFS of 3.7 months [37]. Adding therapies like cetuximab and irinotecan to vemurafenib (known as the VIC regimen) in a phase IB study showed slight improvement, with a median PFS of 7.7 months [38], still far from the desired outcomes. The new standard of care for *BRAF*-mutated mCRC comes from the BEACON study, a phase III trial that randomized 665 pre-treated mCRC patients into three arms: triplet therapy of BRAFi, EGFRi, and MEKi (encorafenib + cetuximab + binimetinib) or the doublet (encorafenib + cetuximab) versus the control arm (investigator’s choice of irinotecan or FOLFIRI + cetuximab) [39]. The study showed clear clinical benefit in both targeted therapy arms compared to the control arm, in patients treated with cetuximab + encorafenib (PFS 4.3 vs. 1.5 months, HR 0.44; OS 9.3 vs. 5.9 months, HR 0.61) and in those treated with cetuximab + encorafenib + binimetinib (PFS 4.5 vs. 1.5 months, HR 0.42; OS 9.3 vs. 5.9 months, HR 0.6). Upon analyzing the results, it was concluded that the doublet had a similar overall efficacy to the triplet. Both regimens improved OS, ORR, and PFS with manageable toxicity (though grade 3 toxicity was slightly higher in the triplet and control arms than in the doublet arm) and similar treatment discontinuation rates. However, the study was not powered sufficiently to formally compare the doublet with the triplet. As a result, cetuximab + encorafenib has become the standard of care for pre-treated patients (not in the first-line setting). This treatment was FDA-approved as of April 2020 [5]. Ongoing studies aim to evaluate the combination of EGFRi, BRAFi, and MEKi in untreated tumors. The ANCHOR study is a phase II trial evaluating the use of encorafenib + binimetinib + cetuximab in the first-line setting. Positive results were obtained, with an ORR of 47.8% and a DCR of 85% (response rates of 50% and stable disease rates of 35%). However, the PFS was disappointing at 5.8 months [40]. Another study exploring different first-line BRAFi options is the BREAKWATER study (phase III), which randomizes 870 patients to receive encorafenib + cetuximab; chemotherapy ± bevacizumab; or chemotherapy + encorafenib + cetuximab [41]. Some preliminary results appear promising, as there is an observed ORR of around 67% and a PFS of approximately 10 months, with data on tolerable toxicity so far [42].

## 5. Mechanisms of Resistance to BRAFi and Therapeutic Options

Despite the promising results shown by BRAFi treatments in combination with other MAPK pathway inhibitors (EGFRi or MEKi) in *BRAF*-mutated metastatic colorectal cancer (mCRC), the emergence of resistance mechanisms to these drugs limits their clinical benefit. While the response rate after double or triple MAPK pathway blockade ranges from 20 to 40% [4], the observed responses are short-lived. This is why resistance mechanisms to BRAFi have been recently studied. They are primarily attributed to the molecular heterogeneity of the tumor. It is common for patients to concurrently present multiple resistance mechanisms [5]. Furthermore, if a cell line is resistant to a specific treatment (BRAFi, MEKi, EGFRi), it will also be resistant to other drugs sharing the same mechanism of action, not just that specific drug (class resistance) [4]. Most of the studies on BRAFi resistance have been conducted based on the prior knowledge in melanoma, although it is not easily extrapolated to colorectal cancer (CCR) due to their distinct origins and molecular biology. BRAFi treatment for melanomas, both in monotherapy and in combination with MEKi, shows a higher response rate compared to mCRC, where monotherapy with vemurafenib does not provide benefit [37]. One of the reasons explaining this difference in response rate is the embryological origin of both types of neoplasms. CCR arises from epithelial cells, which highly express EGFR. Inhibiting only a part of the MAPK pathway (BRAFi or MEKi in monotherapy) triggers a feedback loop that leads to EGFR hyperactivation and subsequent tumor growth [43,44,45]. This phenomenon does not occur in melanoma, which originates from neural crest cells where EGFR is not expressed, thus the feedback loop is not effective. Another hypothesis known as the “big bang” theory [46] suggests that the cellular populations forming the tumor are heterogeneous from the beginning of carcinogenesis. The tumor comprises subpopulations with different mutations (*BRAF* or *KRAS*) at its inception. Therefore, treating with BRAFi eliminates subpopulations with mutations at that level, potentially selecting for a *KRAS*-mutated population resistant to BRAFi. Understanding the MAPK pathway functioning and the physiological negative feedback suppressed by BRAFi monotherapy led to the concurrent administration of drugs inhibiting the pathway at various levels (doublets or triplets with EGFRi and MEKi). Although response rates initially increased, some tumors could escape this drug combination. The analysis of circulating DNA in blood samples from patients included in the BEACON trial revealed that the majority of resistance mechanisms were due to reactivation of the MAPK pathway, despite prior inhibition of BRAF, EGFR, and/or MEK [47]. To better comprehend the underlying resistance mechanisms, Oddo et al. [4] selected cell populations resistant to BRAFi or MEKi monotherapy, but sensitive to different drug combinations (BRAFi + MEKi, BRAFi + EGFRi, MEKi + EGFRi). These cells were exposed to various drugs to generate resistant descendants, aiming to study the molecular mechanisms by which resistance developed and potential therapeutic strategies to overcome it. The drugs included BRAFi (vemurafenib, encorafenib, dabrafenib), MEKi (selumetinib and trametinib), EGFRi (cetuximab), and the PI3K inhibitor apelisib.

### 5.1. Hyperactivation of Other Tyrosine-Kinase Receptors

In cell populations developing resistance to EGFRi, an activation of the MAPK pathway through alternative receptor tyrosine kinases (RTKs) like Her2 or MET was observed [48]. In fact, there are studies demonstrating the clinical benefit of combining BRAF inhibitors (BRAFis) with MET inhibitors (METis) to address this mechanism of resistance [49,50]. Similarly, in BRAFi-resistant melanomas, alternative mechanisms converging on pathway activation were noted, such as overexpression of other RTKs (PDGFR or IGFR-1) [51]. It is also hypothesized that heterogeneous genetic alterations could activate the MAPK pathway alternatively, despite its blockade at different levels.

### 5.2. Feedback Mechanisms and ERK Hyperphosphorylation

Acquired resistance generally results in a retrograde hyperactivation of the MAPK pathway. In the presence of a *BRAF* V600E mutation, the entire pathway downstream is activated, causing a negative feedback loop (upstream) where the final link (ERK) generates a signal that inhibits the pathway. Physiologically, this self-regulates the MAPK pathway, but mutated BRAF proteins escape conventional cellular regulation mechanisms. Administering BRAFi stops ERK activation, disrupting the negative feedback loop. Consequently, EGFR hyperactivation occurs, leading to increased ERK phosphorylation [45]. While there was no difference in the amount of MEK/ERK/AKT between parental and daughter cells, the latter exhibited increased ERK phosphorylation (and sometimes AKT phosphorylation) following combined therapies, while ERK in parental cells remained inhibited [4]. Therefore, increased phosphorylated ERK serves as a mechanism of acquired resistance to BRAFi and could be a potential therapeutic target. ERK inhibitors (ERKis) have been developed for this purpose. Ulixertinib has shown efficacy in tumors with *BRAF* (V600E and non-V600E) and *NRAS* mutations, with low toxicity [52]. Another ERKi that exhibited activity in *BRAF* V600E-mutated mCRC monotherapy is GDC-0994 [53]. Studies also combine ERKi with other targeted therapies. The phase Ib/II HERKULES-3 trial (NCT05039177) combines ERAS-007 (ERKi) with encorafenib and cetuximab in patients with the *BRAF* V600E mutation. Another ongoing study combines a different ERKi (LTT462) with dabrafenib and encorafenib. Concurrent administration of cyclin inhibitors (abemaciclib) with LY3214996, an ERKi, has also been tested (NCT02857270). Additionally, drugs targeting Src homology-2 domain-containing protein tyrosine phosphatase 2 (Shp2), a positive modulator of ERK, have been developed and could mediate acquired resistance. A phase Ib trial combines a Shp2 inhibitor (TNO155) with dabrafenib and trametinib or ERKi [5]. Although this is still preclinical activity, awaiting results, exploring ERK inhibition as a standard practice is worth considering. For now, the question of whether monotherapy or combination with BRAFi is more beneficial remains unresolved, but early results lean towards the combination (enhancing ERKi cytotoxicity) [4].

### 5.3. Structural Modifications of BRAF

The MAPK pathway cascade involves dimerization and subsequent activation of RAF. Despite the extensive clinical use of RAF inhibitors, both dabrafenib and vemurafenib or encorafenib solely inhibit monomeric RAF. An interesting strategy would be to inhibit the dimeric form once it has formed. Pan-RAF inhibitors are being developed for this purpose, such as ponatinib Hybrid Inhibitor 1 (PHI1) [54]. Other therapeutic strategies aim to prevent dimer formation with so-called BRAF paradox breakers. In a phase I/II trial evaluating the paradox breaker PLX8394, an objective response rate (ORR) of 22% was achieved in a population of 45 patients with *BRAF* mutations or fusions [55]. Aberrant splicing generates BRAF forms not sensitive to conventional BRAFi (like *BRAF* V600E DEx), or truncated forms that permanently activate MEK/ERK signaling. Both mechanisms could be involved in BRAFi resistance [51].

### 5.4. Amplifications and Acquired Mutations

When examining the DNA from resistant cells, an increase in the *EGFR* gene copy number was observed in those exposed to the BRAFi + MEKi or BRAFi + EGFRi combinations [4]. While EGFR protein overexpression has been described through immunohistochemistry in cases of primary BRAFi-resistant mutated BRAF CCR [45], gene amplification of *EGFR* has not been established in any study. Other gene amplifications observed in the DNA of resistant cells included *KRAS* and *BRAF* (in cell models subjected to the BRAFi + EGFRi and MEKi + EGFRi combinations, respectively). Amplification of *KRAS* G13D has also been in vitro in melanomas treated with MEKi [56]. These amplifications were determined by PCR and confirmed by FISH. They were found exclusively in the genetic material of resistant cells, absent in parental cells. HER2 or MET amplification was not observed in any case [4]. These findings support the theory that BRAFi resistance mechanisms involve increased activity of other components of the cascade, perpetually keeping the pathway active. Mutations acquired were identified through Sanger sequencing in resistant cell DNA: exons 2, 3, and 4 of *KRAS*; exons 2 and 3 of *NRAS*; exon 15 of *BRAF*; and exons 2 and 3 of *MAP2K1* (gene encoding MEK) [4]. All cells retained the original *BRAF* V600E mutation, while the described acquired mutations were found only in daughter cells. The most frequently observed resistance mechanism was *KRAS* alteration, particularly in exons 2 and 4 (G12D, G13D, A146T/V) [4]. In some cases, different *KRAS* mutations coexisted within the same cell population, suggesting polyclonality. A point mutation was also identified in the EGFR ectodomain (G465R) as an acquired resistance mechanism, enabling ERK phosphorylation and increased cell growth even in the presence of vemurafenib + cetuximab. This mutation had been previously described in BRAF wild-type CCR resistant to cetuximab or panitumumab [57], but had not been observed in mutated *BRAF* tumors until then. These findings highlight the therapeutic utility of inhibiting the MAPK pathway at various vertical levels. This principle is supported by the high response rate observed in melanomas treated with the BRAFi and MEKi dual combination, although results are not fully extrapolatable to mCRC due to the molecular characteristics of each tumor. In tested resistant cells, EGFR overexpression constituted a resistance mechanism to therapeutic doublets (BRAFi + MEKo or + EGFRi), whereas administering a triplet with BRAFi + EGFRi + MEKi restored cell sensitivity to treatment [4]. In another recent study, cells developing resistance to the drug triplet maintained both the *BRAF* V600E mutation and *KRAS* mutation (G12D or G13D) [58], demonstrating that the coexistence of both mutations confers greater treatment resistance to the tumor. However, this condition is very rare, as previously mentioned [15]. Various mechanisms of acquired BRAFi resistance have been described in other tumors, mainly in melanoma, such as the loss of neurofibromin 1 (NF1). NF1 is a tumor suppressor that physiologically inhibits RAS. Its loss results in constant RAS activation of the MAPK pathway, despite targeted therapies. It has been described in 4% of mutated *BRAF* melanomas [59] and plays a significant role in developing BRAFi resistance.

### 5.5. Cell Cycle Dysregulation

Another mechanism involves altered cell cycle regulation. Cyclin D1 binds with CDK4 and CDK6, which phosphorylate retinoblastoma protein, initiating the cell cycle. Cells with hyperactivating cyclin D1 mutations have shown resistance to BRAFi. When this mutation is additionally associated with CDK4 mutations, resistance is further increased [60].

### 5.6. PI3K/AKT/mTOR Pathway Alterations

As reflected in Figure 1, the PI3K/AKT/mTOR pathway converges with the RAS/RAF/MEK/ERK pathway toward the nucleus. In CCR cell samples, PI3K is more activated compared to melanoma cells [61], leading to hypotheses of whether PI3K could be inhibited to bypass BRAFi resistance. Although controversial, current data seem to discount the significant role of the PI3K pathway in developing BRAFi resistance. In a phase II trial conducted with dabrafenib and trametinib, three out of five patients responding to treatment had concurrent *PI3KCA* mutations, with one achieving a complete response. All patients with *PI3K* mutations responded to treatment [62]. In the SWOG S1406 study comparing the VIC regimen (mentioned earlier) and standard chemotherapy with cetuximab, the presence of concurrent PI3K mutations increased progression-free survival (HR of 0.3 vs. 0.6 for PI3K wt) [37]. A comparison between encorafenib + cetuximab and the same doublet with the addition of a PI3K inhibitor (PI3Ki), alpelisib, showed few differences in outcomes: ORR of 29% and 18%, respectively; median PFS of 3.7 months or 4.2 months, respectively, with more toxicity in the alpelisib group [63]. Other hypotheses explored in melanoma regarding BRAFi resistance involve the PI3K/AKT pathway. In cell populations with overexpressed Hepatocyte growth factor (HGF) or its receptor c-MET, hyperactivation of the PI3K/AKT pathway is observed, subsequently activating the RAS/RAF/MEK/ERK pathway. In vitro, sensitivity to HGF or c-MET inhibition was observed, suggesting c-MET could be another therapeutic target [64]. Another therapeutic option explored is inhibiting the PI3K pathway at earlier stages, when it is still interconnected with the MAPK pathway.

### 5.7. Other Potential Mechanisms

Chemokine receptor 4 (CXCR4) is overexpressed in many tumor cells, while its expression is diminished or absent in healthy tissues [65]. CXCR4′s presence in mCRC is associated with increased treatment resistance [66]. Its ligand, CXCL12 or stromal-cell-derived factor-1 (SDF-1), when binding to the receptor, divides it into an alpha subunit (activating the RAS/RAF pathway) and a beta/gamma subunit (activating the PI3K/AKT/mTOR pathway). Hence, crosstalk between both pathways makes CXCR4 a potential new target for BRAFi-resistant mutated *BRAF* tumors. Wnt/beta-catenin pathway involvement in acquired BRAFi resistance has also been observed. BRAF inhibition leads to positive feedback on this pathway [5], promoting cell proliferation, migration, and invasion. Additionally, R-Spondin (RSPO) fusions, which also activate the Wnt/beta-catenin pathway, are significantly associated with *BRAF* mutations, making the combination of BRAFi + Wnti an interesting therapeutic option. A phase I clinical trial for mCRC patients with the *BRAF* V600E mutation combines a Wnti (WNT974) with encorafenib and cetuximab [5]. It has been observed that SRC kinases are systematically activated in *BRAF* V600E CRCs and contribute to increasing tumor growth independently of ERK signaling, through beta-catenin (CTNNB1). The activation of these kinases is facilitated by prostaglandin E2. As we know, the enzyme COX-2 is essential for prostaglandin synthesis, so using a COX-2 inhibitor in combination with BRAF inhibitors (BRAFis) and/or EGFR inhibitors (EGFRis) is an interesting therapeutic strategy that has already shown longer-lasting suppression of tumor growth in patient-derived tumor xenograft models [67]. Furthermore, to enhance the antitumoral activity of BRAFi, VEGFi could be used [68,69], showing long-lasting tumor responses.

## 6. Prognosis and Predictive Biomarkers

As previously mentioned, patients with mutated *BRAF* CRC have a poor prognosis, despite receiving BRAFi. However, there is a subgroup of patients who exhibit a long response to these drugs and better survival. To attempt to define the mentioned prognostic heterogeneity, the BeCool study [70] developed a prognostic scale ranging from 0 to 16 points based on clinical and analytical criteria. It divided patients into low-risk (0–4), intermediate (5–8), and high-risk (9–16) categories, and significant differences were observed in terms of life expectancy in each subgroup (OS of 29.6, 15.5, and 6.6, respectively). It is an easy-to-use tool in clinical practice and could be useful for stratifying patients when enrolling them in future studies.

It is also important to mention the predictive role of the BRAF allele fraction (AF) in patients with *BRAF* V600E CRC who have received treatment with BRAFi + EGFRi ± MEKi. In the prospective cohort designed by Ros et al. [71], it is shown that patients with a high AF (≥2%) experience an increased occurrence of liver metastases and a worse PFS (HR 2.97, 95% CI 1.55–5.69) and OS (HR 3.28, 95% CI 1.58–6.81) compared to patients with a low AF (<2%). Therefore, determining the AF in blood samples can be useful in assessing tumor aggressiveness and identifying the subgroup of patients who may benefit from more intensive treatment.

In the aforementioned BEACON study [39], an attempt was made to establish a correlation between the clinical outcomes and molecular findings in tumor samples. It was concluded that in those BM1 or CMS4 subtypes, there is an increased inflammatory response, as well as an increased ORR in patients receiving the triplet (cetuximab + encorafenib + binimetinib) compared to the doublet (cetuximab + encorafenib): for CMS4, the ORR was 33.3% [95% CI: 21.7–46.7], and for BM1, it was 33.3% [95% CI: 21.4–47.1] in the triplet arm, compared to CMS4: 19.2% [95% CI: 9.6–32.5] and BM1: 14.9% [95% CI: 6.2–28.3] in the doublet arm [72]. Therefore, the molecular subtype of the tumor can serve as a prognostic marker when deciding which patients should receive triple therapy.

As mentioned previously, there is crosstalk between the MAPK pathway and the Wnt/Beta-catenin pathway that can modulate the antitumoral activity of targeted therapy. Recently, it has been observed that patients with MSS *BRAF* V600E CRC with mutations in *RNF43* (a negative regulator of the Wnt pathway) exhibit a better antitumoral response to BRAFi and/or EGFRi than those with RNF43 wild type: they have a longer PFS (HR 0.30; 95% CI 0.12–0.75) and overall survival (OS) (HR 0.26; 95% CI 0.10–0.71). Therefore, RNF43 mutation could be used as a predictive biomarker for treatment response [73,74].

## 7. Conclusions

Mutated *BRAF* mCRC exhibits distinct biological behavior compared to other molecular subtypes of colorectal cancer. While the standard of care for years involved chemotherapy and anti-angiogenic agents, the development of targeted therapy has opened a new avenue for these patients. However, the emergence of BRAFi resistance presents a challenge in providing greater survival and quality of life for these patients. Therefore, continued research into resistance mechanisms and strategies to overcome them is imperative.

## Figures and Tables

**Figure 1 cancers-15-05243-f001:**
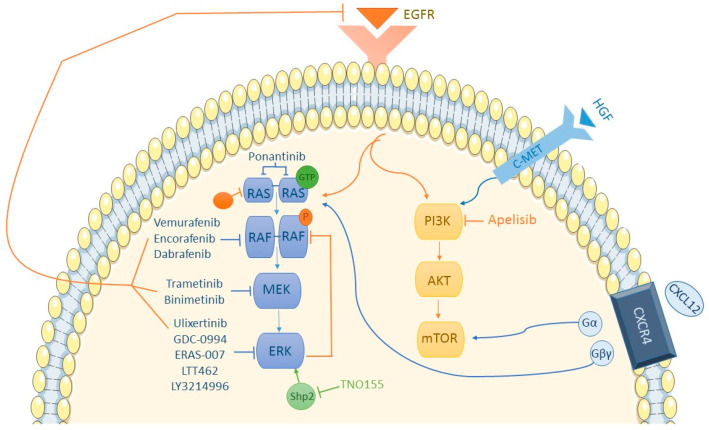
MAPK and PI3K/AKT/mTOR pathway.

## Data Availability

The data presented in this study are available in this article.

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
