# Peer review of "BRAF Inhibitors in Metastatic Colorectal Cancer and Mechanisms of Resistance: A Review of the Literature"

_cancers, 2023, doi:10.3390/cancers15215243_

Round 1

Reviewer 1 Report

Comments and Suggestions for Authors

full and interesting review

-Line 212-213

I would be less strict about the doublet and bevacizumab combination in patients with BRAF mutated tumours. Indeed, Bevacizumab in combination with a triplet or a doublet chemotherapy is considered the best options in patients with BRAF mt tumors. The recent results of the FIRE 4.5 trial promotes the use of FOLFOXIRI plus bevacizumab in all–comer BRAF mt patients eligible for triplet chemotherapy.

I would cite this reference and those of CAIRO 5 in addition to cremolini's meta analysis 

-I would cite international recommendations as ESMO guidelines 2023 for the management of these patients

Author Response

-Line 212-213

I would be less strict about the doublet and bevacizumab combination in patients with BRAF mutated tumours. Indeed, Bevacizumab in combination with a triplet or a doublet chemotherapy is considered the best options in patients with BRAF mt tumors. The recent results of the FIRE 4.5 trial promotes the use of FOLFOXIRI plus bevacizumab in all–comer BRAF mt patients eligible for triplet chemotherapy.

I would cite this reference and those of CAIRO 5 in addition to cremolini's meta analysis 

Thank you for this suggestion, we have amended it

-I would cite international recommendations as ESMO guidelines 2023 for the management of these patients

We have cited the ESMO guidelines as you suggested

Reviewer 2 Report

Comments and Suggestions for Authors

Dear Authors,

I have read your review on the mechanisms of resistance in BRAF mutant CRC treated with BRAF inhibitors with great interest. Given the recent significant advancements in the development of biomarkers within this population, I found your review particularly intriguing. Overall, the review is well-written and easy to read. However, there are important missing data and references necessary for a comprehensive understanding of the issue. Therefore, I would like to suggest adding pertinent literature that has not been mentioned and making some minor changes.

1. Throughout the text, please italicize mutation names when appropriate: BRAF, NRAS, KRAS...

2. Maintain consistency with mutation references throughout the text: BRAF V600E or BRAF-V600E. Avoid using v600E or V600.

3. Figure 1: There are some numbers surrounding the cell membrane; please remove them. Perhaps adding an arrow indicating the negative feedback through the EGFR receptor would be appreciated by the readers.

4. In Point 2.2, you can summarize the content (e.g., perhaps the information about the trials evaluating EGFR inhibitors can be more concise). In the same point, the work from Barras et al., published in Clinical Cancer Research, mentions "BM1" and "BM2." Please modify "BMS-1" and "BMS-2" as these are not accurate abbreviations. Additionally, in Point 2.2, you should emphasize that almost 60-70% of BRAF CRC tumors belong to CMS1, as this fact may explain the successful outcomes when combining BRAF inhibitors with immunotherapy (Van Morris et al., ASCO 2022, and Ryan Corcoran, Nature Medicine 2023).

5. In Point 3: Patients with prolonged QTc should not receive BRAF inhibitors, as they can prolong QTc (toxicity class). I recommend deleting this sentence as it falls outside the scope of the review.

6. In Point 4, move Point 4.2 to the first position, and summarize all the data related to chemotherapy and EGFR inhibitors. The key message here is that clinical outcomes are poor in this population, and EGFR inhibitors are not recommended. Please refer to the ESMO guidelines:

- Cervantes A, Adam R, Roselló S, Arnold D, Normanno N, Taïeb J, Seligmann J, De Baere T, Osterlund P, Yoshino T, Martinelli E; ESMO Guidelines Committee. Electronic address: [email protected]. Metastatic colorectal cancer: ESMO Clinical Practice Guideline for diagnosis, treatment and follow-up. Ann Oncol. 2023 Jan;34(1):10-32. doi: 10.1016/j.annonc.2022.10.003. Epub 2022 Oct 25. PMID: 36307056.

Afterward, Point 4.1 (now 4.2) and 4.3 will fit better. When discussing the BREAKWATER trial, consider adding a brief sentence referring to the presented results to help readers understand the significance of this trial, which may be practice-changing (Tabernero J et al., ESMO 2022, oral presentation).

7. In Point 5, when discussing the negative feedback through the EGFR receptor, refer to two papers:

- Prahallad A, Sun C, Huang S, et al. Unresponsiveness of colon cancer to BRAF(V600E) inhibition through feedback activation of EGFR. Nature. 2012 Mar 26;483(7387):100–103

- Corcoran RB, Ebi H, Turke AB, et al. EGFR-mediated re-activation of MAPK signaling contributes to insensitivity of BRAF mutant color-ectal cancers to RAF inhibition with vemurafenib. Cancer Discov. 2012 Mar;2(3):227–235

Additionally, mention a pivotal paper summarizing the mechanism of resistance in the BEACON trial:

- Kopetz S, Murphy DA, Pu J, et al. Genomic mechanisms of acquired resistance of patients (pts) with BRAF V600E-mutant (mt) metastatic colorectal cancer (mCRC) treated in the BEACON study. Ann Oncol. 2022 Sep;33:S681–2.

8. In Point 5.1, concerning MET amplification as an acquired mechanism of resistance, include two relevant papers demonstrating that acquired MET amplification can be targeted with BRAF inhibitors + MET inhibitors. It may not be necessary to provide extensive details, but highlight the successful blockade of MET + BRAF, which is clinically feasible:

- Pietrantonio F, Oddo D, Gloghini A, et al. MET-driven Resistance to Dual EGFR and BRAF blockade may be overcome by switching from EGFR to MET inhibition in BRAF-Mutated colorectal cancer. Cancer Discov. 2016 Sep 1;6(9):963–971.

- Ros J, Elez E. Overcoming acquired MET amplification after encorafenib-cetuximab in BRAF-V600E mutated colorectal cancer. Eur J Cancer. 2022 Sep;172:326–328. doi: 10.1016/j.ejca.2022.06.026

In Point 5.7, briefly mention several papers (perhaps 2-3 lines each). Firstly, include a recently published paper on COX inhibitors to enhance antitumor activity, as well as a recent review on biomarkers in BRAF inhibitors:

- Ruiz-Saenz A, Atreya CE, Wang C, et al. A reversible SRC-relayed COX2 inflammatory program drives resistance to BRAF and EGFR inhibition in BRAFV600E colorectal tumors. Nat Cancer. 2023 Feb 9;4(2):240–256.

In the same point, highlight that anti-VEGF blockade added to BRAF inhibitors may enhance BRAF inhibitor activity:

- Bottos A, Martini M, Di Nicolantonio F, et al. Targeting oncogenic serine/threonine-protein kinase BRAF in cancer cells inhibits angiogenesis and abrogates hypoxia. Proc Natl Acad Sci. 2012 Feb 7;109(6):E353–9.

- Comunanza V, Corà D, Orso F, et al. VEGF blockade enhances the antitumor effect of BRAFV600E inhibition. EMBO Mol Med. 2017 Feb 14;9(2):219–237.

Importantly, the conclusion section should be point 7, and point 6 should refer to prognostic and predictive biomarkers, which will be important in the review. To fully understand the significance of resistance after BRAF inhibitor  treatment, prognostic and predictive biomarkers should be summarized and mentioned.

Patients with BRAF mutant colorectal cancer (CRC) have poor prognostic outcomes regardless of receiving a BRAF inhibitor. However, prognostic factors have demonstrated that some patients will experience long-lasting survival or prolonged responses to BRAF inhibitors. Therefore, several papers should be mentioned to explain the clinical and molecular heterogeneity of this particular subgroup:

- Prognostic: The BeCool score and the prognostic value of BRAF allele fraction should be highlighted. Also, the molecular correlates from the BEACON trial should be mentioned:

- Loupakis F, Intini R, Cremolini C, et al. A validated prognostic classifier for V600EBRAF-mutated metastatic colorectal cancer: the 'BRAF BeCool' study. Eur J Cancer. 2019 Sep;118:121-130. doi: 10.1016/j.ejca.2019.06.008. Epub 2019 Jul 19. PMID: 31330487.

- Ros J, Matito J, Villacampa G, et al. Plasmatic BRAF-V600E allele fraction as a prognostic factor in metastatic colorectal cancer treated with BRAF combinatorial treatments. Ann Oncol. 2023 Mar;34 (6):543–552

- Kopetz S, Murphy DA, Pu J, et al. Molecular correlates of clinical benefit in previously treated patients (pts) with BRAF V600E-mutant metastatic colorectal cancer (mCRC) from the BEACON study. J Clin Oncol. 2021 May 20;39(15_suppl):3513–3513

- Predictive: Several papers have demonstrated the predictive value of RNF43 mutations. As the authors state in the review, APC and WNT pathways are relevant steps in the carcinogenesis process of colorectal cancer. The authors have already introduced the WNT pathway when discussing RSPO. These papers suggest that there is crosstalk between the MAPK pathway (BRAF) and the WNT pathway (RNF43):

- Elez E, Ros J, Fernández J, et al. RNF43 mutations predict response to anti-BRAF/EGFR combinatory therapies in BRAFV600E metastatic colorectal cancer. Nat Med. 2022 Sep 12;28(10):2162–2170. 

- Quintanilha JCF, Graf RP, Oxnard GR. BRAF V600E and RNF43 co-mutations predict patient outcomes with targeted therapies in real-world cases of colorectal cancer. Oncology. 2023 Feb 13;28(3): e171–e174

Author Response

1.Throughout the text, please italicize mutation names when appropriate: BRAF, NRAS, KRAS...

Thank you for this suggestion, we have amended it

2. Maintain consistency with mutation references throughout the text: BRAF V600E or BRAF-V600E. Avoid using v600E or V600.

Thank you for this suggestion, we have amended it

3. Figure 1: There are some numbers surrounding the cell membrane; please remove them. Perhaps adding an arrow indicating the negative feedback through the EGFR receptor would be appreciated by the readers.

Thank you for this suggestion, I have added the arrow indicating the negative feedback throug EGFR. I can not longer see the numbers surrounding the cell membrane

4. In Point 2.2, you can summarize the content (e.g., perhaps the information about the trials evaluating EGFR inhibitors can be more concise). In the same point, the work from Barras et al., published in Clinical Cancer Research, mentions "BM1" and "BM2." Please modify "BMS-1" and "BMS-2" as these are not accurate abbreviations. Additionally, in Point 2.2, you should emphasize that almost 60-70% of BRAF CRC tumors belong to CMS1, as this fact may explain the successful outcomes when combining BRAF inhibitors with immunotherapy (Van Morris et al., ASCO 2022, and Ryan Corcoran, Nature Medicine 2023)

Thank you for this remark. We have modified it as you suggested.

5. In Point 3: Patients with prolonged QTc should not receive BRAF inhibitors, as they can prolong QTc (toxicity class). I recommend deleting this sentence as it falls outside the scope of the review.

Thank you for this suggestion, we have deleted it.

6. In Point 4, move Point 4.2 to the first position, and summarize all the data related to chemotherapy and EGFR inhibitors. The key message here is that clinical outcomes are poor in this population, and EGFR inhibitors are not recommended. Please refer to the ESMO guidelines:

- Cervantes A, Adam R, Roselló S, Arnold D, Normanno N, Taïeb J, Seligmann J, De Baere T, Osterlund P, Yoshino T, Martinelli E; ESMO Guidelines Committee. Electronic address: [email protected]. Metastatic colorectal cancer: ESMO Clinical Practice Guideline for diagnosis, treatment and follow-up. Ann Oncol. 2023 Jan;34(1):10-32. doi: 10.1016/j.annonc.2022.10.003. Epub 2022 Oct 25. PMID: 36307056.

Afterward, Point 4.1 (now 4.2) and 4.3 will fit better. When discussing the BREAKWATER trial, consider adding a brief sentence referring to the presented results to help readers understand the significance of this trial, which may be practice-changing (Tabernero J et al., ESMO 2022, oral presentation).

Thank you for this suggestion. We have summarized all the data related to EGFR inhibitors and chemotherapy. We have cited ESMO guidelines and BREAKWATER results

7. In Point 5, when discussing the negative feedback through the EGFR receptor, refer to two papers:

- Prahallad A, Sun C, Huang S, et al. Unresponsiveness of colon cancer to BRAF(V600E) inhibition through feedback activation of EGFR. Nature. 2012 Mar 26;483(7387):100–103

- Corcoran RB, Ebi H, Turke AB, et al. EGFR-mediated re-activation of MAPK signaling contributes to insensitivity of BRAF mutant color-ectal cancers to RAF inhibition with vemurafenib. Cancer Discov. 2012 Mar;2(3):227–235

Additionally, mention a pivotal paper summarizing the mechanism of resistance in the BEACON trial:

  • Kopetz S, Murphy DA, Pu J, et al. Genomic mechanisms of acquired resistance of patients (pts) with BRAF V600E-mutant (mt) metastatic colorectal cancer (mCRC) treated in the BEACON study. Ann Oncol. 2022 Sep;33:S681–2.

Thank you for this suggestion. We have cited the papers you mentioned.

8. In Point 5.1, concerning MET amplification as an acquired mechanism of resistance, include two relevant papers demonstrating that acquired MET amplification can be targeted with BRAF inhibitors + MET inhibitors. It may not be necessary to provide extensive details, but highlight the successful blockade of MET + BRAF, which is clinically feasible:

- Pietrantonio F, Oddo D, Gloghini A, et al. MET-driven Resistance to Dual EGFR and BRAF blockade may be overcome by switching from EGFR to MET inhibition in BRAF-Mutated colorectal cancer. Cancer Discov. 2016 Sep 1;6(9):963–971.

- Ros J, Elez E. Overcoming acquired MET amplification after encorafenib-cetuximab in BRAF-V600E mutated colorectal cancer. Eur J Cancer. 2022 Sep;172:326–328. doi: 10.1016/j.ejca.2022.06.026

In Point 5.7, briefly mention several papers (perhaps 2-3 lines each). Firstly, include a recently published paper on COX inhibitors to enhance antitumor activity, as well as a recent review on biomarkers in BRAF inhibitors:

- Ruiz-Saenz A, Atreya CE, Wang C, et al. A reversible SRC-relayed COX2 inflammatory program drives resistance to BRAF and EGFR inhibition in BRAFV600E colorectal tumors. Nat Cancer. 2023 Feb 9;4(2):240–256.

In the same point, highlight that anti-VEGF blockade added to BRAF inhibitors may enhance BRAF inhibitor activity:

- Bottos A, Martini M, Di Nicolantonio F, et al. Targeting oncogenic serine/threonine-protein kinase BRAF in cancer cells inhibits angiogenesis and abrogates hypoxia. Proc Natl Acad Sci. 2012 Feb 7;109(6):E353–9.

- Comunanza V, Corà D, Orso F, et al. VEGF blockade enhances the antitumor effect of BRAFV600E inhibition. EMBO Mol Med. 2017 Feb 14;9(2):219–237.

Importantly, the conclusion section should be point 7, and point 6 should refer to prognostic and predictive biomarkers, which will be important in the review. To fully understand the significance of resistance after BRAF inhibitor  treatment, prognostic and predictive biomarkers should be summarized and mentioned.

Patients with BRAF mutant colorectal cancer (CRC) have poor prognostic outcomes regardless of receiving a BRAF inhibitor. However, prognostic factors have demonstrated that some patients will experience long-lasting survival or prolonged responses to BRAF inhibitors. Therefore, several papers should be mentioned to explain the clinical and molecular heterogeneity of this particular subgroup:

- Prognostic: The BeCool score and the prognostic value of BRAF allele fraction should be highlighted. Also, the molecular correlates from the BEACON trial should be mentioned:

- Loupakis F, Intini R, Cremolini C, et al. A validated prognostic classifier for V600EBRAF-mutated metastatic colorectal cancer: the 'BRAF BeCool' study. Eur J Cancer. 2019 Sep;118:121-130. doi: 10.1016/j.ejca.2019.06.008. Epub 2019 Jul 19. PMID: 31330487.

- Ros J, Matito J, Villacampa G, et al. Plasmatic BRAF-V600E allele fraction as a prognostic factor in metastatic colorectal cancer treated with BRAF combinatorial treatments. Ann Oncol. 2023 Mar;34 (6):543–552

- Kopetz S, Murphy DA, Pu J, et al. Molecular correlates of clinical benefit in previously treated patients (pts) with BRAF V600E-mutant metastatic colorectal cancer (mCRC) from the BEACON study. J Clin Oncol. 2021 May 20;39(15_suppl):3513–3513

- Predictive: Several papers have demonstrated the predictive value of RNF43 mutations. As the authors state in the review, APC and WNT pathways are relevant steps in the carcinogenesis process of colorectal cancer. The authors have already introduced the WNT pathway when discussing RSPO. These papers suggest that there is crosstalk between the MAPK pathway (BRAF) and the WNT pathway (RNF43):

- Elez E, Ros J, Fernández J, et al. RNF43 mutations predict response to anti-BRAF/EGFR combinatory therapies in BRAFV600E metastatic colorectal cancer. Nat Med. 2022 Sep 12;28(10):2162–2170. 

  • Quintanilha JCF, Graf RP, Oxnard GR. BRAF V600E and RNF43 co-mutations predict patient outcomes with targeted therapies in real-world cases of colorectal cancer. Oncology. 2023 Feb 13;28(3): e171–e174

Thank you for this suggestion, we have corrected it

Round 2

Reviewer 2 Report

Comments and Suggestions for Authors

Dear authors, 

Thank you for your corrections, 

Great job